# Development of C_4_ Biochemistry and Change in Expression of Markers for Photosystems I and II in the Single-Cell C_4_ Species, *Bienertia sinuspersici*

**DOI:** 10.3390/plants12010077

**Published:** 2022-12-23

**Authors:** Makoto Yanagisawa, Simon D. X. Chuong

**Affiliations:** 1Departments of Botany and Plant Pathology, Purdue University, West Lafayette, IN 47907, USA; 2Department of Biology, University of Waterloo, Waterloo, ON N2L 3G1, Canada

**Keywords:** single-cell C_4_ photosynthesis, rubisco large subunit, pyruvate Pi dikinase

## Abstract

*Bienertia sinuspersici* is one of four identified terrestrial plants that perform C_4_ photosynthesis within a single chlorenchyma cell via the compartmentation of organelles and photosynthetic enzymes. The patterns of accumulation of key photosynthetic enzymes and transcripts in developing leaves were examined using immunolocalization and *in situ* hybridization. The polypeptides of Rubisco large subunit (RbcL) and pyruvate Pi dikinase (PPDK) accumulated equally in all chloroplasts before the formation of two intracellular cytoplasmic compartments: the central (CCC) and peripheral (PCC) cytoplasmic compartments. The differential accumulation of these enzymes was not completed until the leaf had reached maturity, indicating that the transition from C_3_ to C_4_ photosynthesis occurred during leaf maturation. In mature chlorenchyma cells, RbcL accumulated 20-fold higher in the CCC than in the PCC, while PPDK exhibited a concentration gradient that was the lowest in the chloroplasts in the central region of the CCC and the highest in PCC chloroplasts. The pattern of *rbcL* transcript accumulation followed that of its polypeptides in developing leaves, suggesting that the expression of this gene was likely controlled by transcriptional and/or post-transcriptional processes. Immunocytochemical results examining the distribution of photosystems I and II in the chloroplasts of chlorenchyma cells from mature leaves showed that PSII is more abundant in chloroplasts of the central compartment, whereas PSI is higher in those of the peripheral compartment. The quantitative real-time PCR results of *rbcL*, *psbA*, and *psaB* transcripts from the isolated chloroplasts of each compartment further supported this observation. Our results suggest that multiple levels of regulation play a role in controlling the differential accumulation of photosynthetic gene expression in the dimorphic chloroplasts of single-cell C_4_ species during leaf development.

## 1. Introduction

Relative to C_3_ plants, C_4_ plants have several advantages, such as a high CO_2_ assimilation rate, a low photorespiration rate, and high nitrogen and water use efficiency in hot and dry environments [1]. While C_3_ plants have only the mesophyll photosynthetic cell type in the leaves, C_4_ plants have distinct mesophyll and bundle sheath cell types for photosynthesis. This dual-cell system in typical C_4_ leaves is known as the Kranz anatomy. Kranz-less exceptions in terrestrial plants are found in four species of the Chenopodiaceae family, which can perform C_4_ photosynthesis in individual chlorenchyma cells [2,3,4,5,6]. C_4_ photosynthesis in single-cell type C_4_ species is achieved with the spatial compartmentation of organelles and key C_4_ enzymes into two distinct cytoplasmic compartments. *Bienertia sinuspersici*, *B. cycloptera,* and *B. kaverense* have a similar cell anatomy and single-cell type C_4_ mechanisms, with all having a central cytoplasmic compartment (CCC) and a peripheral cytoplasmic compartment (PCC) that function analogous to bundle sheath and mesophyll cells, respectively, in Kranz-type C_4_ species [3,6,7,8]. The CCC is enriched in mitochondria and chloroplasts with a high granal index, whereas the PCC lacks mitochondria and has agranal chloroplasts. In the *Bienertia* C_4_ photosynthetic system, the initial fixation of atmospheric CO_2_ occurs in the cytoplasm of the PCC by phosphoenolpyruvate carboxylase (PEPC), and the C_4_ acids produced are transported to the CCC through transvacuolar cytoplasmic channels. The decarboxylation of C_4_ acids is processed by NAD-malic enzymes (MEs) in mitochondria in the CCC, and the released CO_2_ is refixed by ribulose 1,5-bisphosphate carboxylase/oxygenase (Rubisco) in the chloroplasts surrounding the mitochondria in the CCC. C_3_ byproducts are shuttled back to the chloroplasts in the PCC, where enriched pyruvate Pi dikinase (PPDK) regenerates PEP [2,8,9].

For terrestrial plants, the spatial compartmentation of enzymes is essential to perform C_4_ photosynthesis in both Kranz-type and single-cell type anatomies. However, the timing of enzyme partitioning in different cells or compartments during leaf development varies among C_4_ species. In the C_4_ maize monocot, *Rubisco* transcripts accumulate in the earliest developmental stage, although its polypeptides are undetectable [10]. In contrast, the accumulation of *PEPC* and *NADP-ME* transcripts coincides with the accumulation of their corresponding polypeptides [11]. Moreover, their mRNAs and proteins accumulate after the differentiation of the mesophyll and bundle sheath cells, and are localized in a cell-specific manner. *PPDK* transcripts accumulate mostly in mesophyll cells, but are also detectable in the bundle sheath cells of mature maize leaves [12]. In the C_4_ dicot *Amaranthus*, both the transcripts and polypeptides of the Rubisco large subunit (RbcL) are detected in the initial stage of leaf development [13,14]. However, the transcripts and polypeptides of Rubisco are localized in both bundle sheath and mesophyll cells. *PEPC* and *PPDK* transcripts are abundant in the leaf primordia, where these polypeptides are not detected [15]. The non-cell-specific accumulation of these transcripts continues in later developmental stages, while their polypeptides increase and show cell-specific accumulation. These studies suggest that cell-specific localization is regulated independently among the photosynthetic enzymes and also among species. Moreover, the cell-specific gene expression of these enzymes is differentially regulated during leaf development; in the earlier stage, it is regulated at post-transcriptional levels, and in the later stage, it is regulated at transcriptional levels [10,11,16,17,18,19,20,21,22,23]. 

Immunolocalization and *in situ* hybridization are powerful methods to determine the distribution of polypeptides and transcripts, and have been used extensively in studies on Kranz-type C_4_ plants [24,25]. The immunolocalization technique was extensively and successfully applied to the single-cell type C_4_ species in detecting the C_4_ enzyme distribution [2,3,7,8,26,27]. However, these immunological studies mainly analyzed the enzyme distribution at the light microscopic level. High-resolution analysis at the electron microscopic level is especially important in the study of single-cell type C_4_ species because enzyme compartmentation occurs at the subcellular level, unlike in Kranz-type C_4_ plants that utilize two cell types. The determination of transcript distribution using *in situ* hybridization in single-cell type C_4_ species is more challenging than that for Kranz-type C_4_ plants because transcripts of nuclear-encoding genes such as *PEPC*, *PPDK*, and *NAD-ME* are localized in the cytoplasm of a single chlorenchyma cell. Therefore, we decided to focus exclusively on the transcript localization of chloroplast-encoding photosynthetic genes, including *rbcL*, *psaB*, and *psbA*.

In this study, the relative abundance of photosynthetic polypeptides was examined via the transmission electron microscopic (TEM) analysis of immunolocalization in developing leaves of *B. sinuspersici*. The distribution and accumulation of plastid transcripts were investigated with *in situ* hybridization and real-time qPCR. Our results indicate that the differential accumulation of Rubisco and PPDK polypeptides in dimorphic chloroplasts progressively occurs during leaf maturation in this single-cell C_4_ species. Furthermore, the distribution of *rbcL* transcripts appears to be regulated at the transcriptional and/or post-transcriptional levels.

## 2. Results

### 2.1. Quantitative Immunolocalization Analysis of RbcL and PPDK in the Developing Leaves of Bienertia sinuspersici

The leaves of *B. sinuspersici* were divided into five developmental stages: youngest (0.1 cm), young (0.3 cm), intermediate (0.5–0.6 cm), premature (1.0–1.2 cm), and mature (>2 cm) (Figure 1A). In the youngest leaves, chlorenchyma cells are mainly occupied by nuclei and cytoplasm (Figure 1B). There are several chloroplasts of similar morphology and small vacuoles in the cytoplasm. In young leaves, large vacuoles begin to develop and fuse together, pressing the cytoplasm and chloroplasts either adjacent to the nucleus or along the cell wall (Figure 1C). In intermediate leaves, the development of a unique single-cell C_4_ feature is detected, as organelles begin to accumulate adjacent to the nucleus, forming the central cytoplasmic compartment (CCC) in the center of cell (Figure 1D). The CCC is connected to the peripheral cytoplasmic compartment (PCC) by numerous cytoplasmic strands. The characteristic feature of single-cell C_4_ is fully established in the chlorenchyma cells of premature and mature leaves, along with morphologically distinct chloroplasts containing CCC and PCC (Figure 1E,F). Chloroplasts in the CCC have numerous well-developed grana (Figure 1G), whereas those of the PCC have less-developed grana, interconnected by numerous long intergranal thylakoids (Figure 1H). Mitochondria are also abundant in the CCC compartment, as demonstrated by the immunolocalization of glycine decarboxylase specifically within these organelles (Figure 1G). The identity of the organelles that are often found associating with chloroplasts in the PCC was confirmed as peroxisomes using immunolocalization of catalase protein (Figure 1H).

To examine the relative abundance of Rubisco and PPDK at different developmental stages at a higher resolution, we performed immunolocalization analysis for Rubisco large subunit (RbcL) and PPDK, and then observed specific reactions to the polypeptides using the TEM analysis of the density of gold particles. Control experiments using preimmune rabbit sera or the removal of the primary antibody showed minimal or no background labeling (data not shown). For TEM analysis, chloroplasts in the CCC were further divided into two groups: chloroplasts in the inner (CCC inner: CCCI chloroplasts) and outer (CCC outer: CCCO chloroplasts) regions of the CCC. A low level of RbcL labelling was observed in the cells of the youngest leaves, and it gradually increased in all chloroplast types until the premature stage (Figure 2A–H and Figure 3). RbcL accumulated evenly in chloroplasts, with a reduced amount in the PCC of intermediate and premature leaves (Figure 2C–H and Figure 3). From premature to mature leaves, the amount of RbcL was drastically reduced in PCC chloroplasts, while it increased in CCCI and CCCO chloroplasts (Figure 2F–K and Figure 3). The density of gold particles for RbcL in both CCCI and CCCO chloroplasts was more than 20-fold higher compared to that of PCC chloroplasts in mature leaves (Figure 4). In contrast to RbcL, the labelling of PPDK was rarely observed in youngest leaves (Figure 3A), and accumulated in young leaves, with the density remaining unchanged until the intermediate stage (Figure 3B–E and 3). In premature leaves, PPDK started showing a concentration gradient with a low density in the CCCI, moderate in the CCCO, and high in PCC chloroplasts (Figure 3F–H and 3). This gradient became more distinct in mature leaves, showing approximately 2- and 6-fold more PPDK accumulation in the PCC than that in the CCCO and the CCCI, respectively (Figure 3I–K and 3).

### 2.2. Quantitative Immunolocalization Analysis of PsbO and Cytochrome f in Mature Leaves 

Previously, Voznesenskaya et al. (2002) reported that the granal index is 1.5-fold higher in CCC than that in the PCC chloroplasts of *B. cycloptera.* [3] Therefore, it was speculated that Photosystem II (PSII) would accumulate less, and its activity would be lower in PCC than that in CCC chloroplasts. However, a study showed that isolated PCC chloroplasts had a similar ability of photosynthetic oxygen evolution to that of CCC chloroplasts in *B. sinuspersici* [28]. Moreover, photosystem (PS) proteins are differentially distributed in the mesophyll and the bundle sheath cell in Kranz-type C_4_ species [29,30,31]. To investigate PSI and PSII distribution and accumulation in B. sinuspersici, we performed the immunolocalization of PsaB (PSI), PsbA (PSII), PsbO, which is an 80 kD protein in the oxygen evolving complex associated with the PS II reaction center. The distribution of cytochrome f in the two types of chloroplasts was also compared with the PSI and PSII distribution, because cytochrome f is similarly distributed between mesophyll and bundle sheath chloroplasts in a Kranz-type C_4_ maize species [30]. The gold-labelled PsbO was observed mostly on the grana membrane (Figure 5A,B and insets), whereas cytochrome f distributed evenly to the grana membrane and stroma lamellae as expected (Figure 5D,E, and insets). The density of PsbO in the CCCI and CCCO chloroplasts was 1.6-fold higher than that in the PCC chloroplasts (Figure 5C), while the density of cytochrome f labeled gold particles was not significantly lower in the CCCI chloroplasts compared to the PCC and CCCO chloroplasts (Figure 5F). Similarly, the distribution of the PsbA subunits of PSII was more than 2-fold higher in the CCC chloroplasts than that of the PCC chloroplasts (Figure 5G–I and insets). However, the distribution of the PsaB subunits was more abundant in the PCC chloroplasts compared to that of the CCC chloroplasts (Figure 5J–L and insets).

### 2.3. Subcellular Localization and Quantification of Chloroplast-Encoded Photosynthetic Transcripts

Photosynthetic proteins are differentially distributed in dimorphic chloroplasts in mature leaves of *Bienertia* as previously reported [7,8] and as found in our immunolocalization studies. Therefore, the distributions of their transcripts and proteins were expected to correlate. Because nuclear-encoded transcripts such as PPDK were localized in the cytosol and were difficult to resolve with *in situ* hybridization at the light microscopic level, plastid-encoded transcripts were chosen for further analysis. The subcellular localizations of *rbcL*, *psaB*, *psbA*, and 16S rRNA in the developing leaves of *B. sinuspersici* were determined by *in situ* hybridization (Figure 6). Leaf sections were counter-stained with Safranin O after color development for the better visualization of cell structures. Sense strand *rbcL* RNA was hybridized to the leaf sections as a negative control, which only showed orange, pink, and red as a result of Safranin O staining (Figure 6A–C). The nucleus is the largest structure in the young leaves (Figure 6, left panels), while the CCC forms and becomes the largest structure in intermediate and mature leaves (Figure 6, middle and right panels). The rbcL transcripts accumulated in all scattered chloroplasts (purple dots) in young leaves where the CCC had not yet formed (Figure 6D). In intermediate leaves, abundant *rbcL* mRNA was observed both in the CCC and PCC chloroplasts (Figure 6E). However, the majority of *rbcL* transcripts were localized in the CCC with only a low level of accumulation in the PCC chloroplasts in the mature leaves (Figure 6F). This observation correlates with the results of Koteyeva et al. (2016), who showed similar plastid-specific accumulation of rbcL transcripts along the longitudinal leaf gradients of *Bienertia sinuspersici* [32]. The PCC chloroplasts were barely visible without Safranin O counterstaining (Figure 6F inset). In contrast to our expectation, psaB (Figure 6G–I) and psbA (Figure 6J–L) transcripts exhibited a similar distribution to that of the *rbcL* transcripts. The 16S rRNA accumulated in similar amounts in chloroplasts of both the CCC and PCC in all leaf stages indicating that this chloroplastic rRNA can be utilized as an internal control for the quantification of transcripts (Figure 6M–O). 

Overall, the *in situ* hybridization analysis showed that *rbcL*, *psaB*, and *psbA* transcripts accumulated similarly in the two types of chloroplasts in young and intermediate leaves, whereas they were more abundant in the CCC than that in the PCC in mature leaves (Figure 5D–L). This is further supported by the quantitative real-time PCR results (Figure 6). These results indicate that the expression of these chloroplastic genes is at least partially controlled by transcription and/or mRNA stability during leaf development. In particular, the distribution of *rbcL* transcripts appeared to correlate with RbcL protein distribution throughout leaf development, suggesting that rbcL expression is predominantly controlled at these levels.

## 3. Discussion

In the youngest leaves, chlorenchyma cells were mainly occupied by nuclei and the cytoplasm (Figure 1A). There were several undifferentiated chloroplasts that were similar in morphology and small vacuoles in the cytoplasm. In the young leaves, vacuoles began to fuse together, pressing cytoplasm and chloroplasts either adjacent to the nucleus or along the cell wall (Figure 1B). In the intermediate leaves, there is evidence for the formation of the CCC in the center of cells (Figure 1C). Premature leaves showed chlorenchyma cells with a directional development (Figure 1E). In mature leaves, chlorenchyma cells were fully expanded, containing CCC and PCC chloroplasts morphologically distinct from each other (Figure 1F). In general, chloroplasts in the CCC had well-developed grana, and particularly those in the outer layer of the CCC contained large starch grains [7]. Mitochondria were also abundantly packed in this compartment (Figure 1G). In previous studies, some mitochondrial-like structures were observed in the PCC associating with chloroplasts by using rhodamine 123 staining, although the majority of mitochondria were localized in the CCC in the mature chlorenchyma cell of *B. sinuspersici* [7,8,33]. However, at the electron microscopic level, no mitochondrion was found in the PCC. Instead, approximately 20% (20.1% ± 6%) of the PCC chloroplasts were associated with peroxisome-like organelles in the ultrathin sections (60–80 nm) of mature leaves (Figure 1H). 

In the monocot C_4_ maize and dicot C_4_ *Atriplex rosea*, C_4_ enzymes and Rubisco accumulate in a cell-specific manner throughout development in light-grown leaves [17,34]. In addition, the *rbcL* and Rubisco small subunit (*RbcS*) transcripts accumulate before cell differentiation in maize [35]. In contrast, in another dicot C_4_ *Amaranthus hypochondriacus*, RbcL and RbcS mRNA and proteins occur in both mesophyll and bundle sheath cells in 5 mm long leaves and the cell-specific accumulation of Rubisco occurs in 10 mm long leaves [13]. Wang et al. (1992) showed that RbcL and RbcS levels are quickly reduced in mesophyll cells and accumulate only in bundle sheath cells within 24 to 36 h, as the leaves develop from 5 to 10 mm in length [13]. Our results indicate that the compartmentation of RbcL and PPDK accumulates progressively during the late stage of development, between premature (1.0–1.2 cm in length) and mature (>2 cm in length) leaves of *B. sinuspersici* (Figure 4). Notably, mature leaves of *B. sinuspersici* expand to 2–3 cm length, while the fully expanded leaves of *A. hypochondriacus* reach 10 cm. This indicates that, in *B. sinuspersici*, unlike some Kranz type C_4_ plants, a default C_3_-like mode of photosynthesis appears to be maintained in early stages of leaf development with the full C_4_ mode occurring in the later stages of leaf maturation. 

Although it is currently unclear how the differentiation of the two types of chloroplasts occur within the same cell, Offerman et al. (2011) reported that isolated PCC chloroplasts showed 4- to 5-fold more PPDK than that of CCC chloroplasts [28]. However, our results did not indicate this much difference because of the moderate accumulation of PPDK in CCCO chloroplasts (Figure 4). The discrepancy between these results might be attributed to differences in growth conditions such as the light intensity (1000 vs. 350 µmol s^−1^ m^−2^) and temperature (day/night: 35/18 vs. 25/18 °C), which probably affected the overall biochemical processes in this species. Similar discrepancies were also observed in δ^13^C analyses in studies showing that *B. sinuspersici* plants from natural habitats have –13.8‰ suggestive of C_4_ photosynthesis, whereas plants grown in a growth chamber under conditions of approximately 400 µmol s^−1^ m^−2^ light intensity and 25/15 °C day/night temperatures have –17.4‰ indicating C_4_-like photosynthesis [3,5,27,36,37]. Freitag and Stichler (2002), and Edwards et al. (2004) suggested that the plasticity of the photosynthetic mode in *Bienertia* species depends on growth conditions [4,9]. Moreover, a recent study showed that transit peptide elements play an important role in the selective protein targeting, resulting in the differential accumulation of PPDK in the dimorphic chloroplasts [38]. Together with our data, the transition from a default C_3_-like mode to a full C_4_ mode of photosynthesis is likely regulated not only developmentally but also environmentally in *B. sinuspersici*, thereby potentially affecting the distribution of C_4_ enzymes including PPDK.

The distribution of light harvesting complex II and PSII affects the development of grana stacks [39,40]. For instance, maize develops dimorphic chloroplasts; mesophyll chloroplasts have well-developed grana, while bundle sheath chloroplasts are deficient in grana [9]. Proteomic analyses in maize revealed that the content of PSII in bundle sheath cells is 55% lower than that in mesophyll cells [29]. Furthermore, a large-scale proteomic study by Offermann et al. (2015) revealed that most photosystem II (PSII) proteins were more abundant in chloroplasts of the central compartment than those of the peripheral compartment [41]. In this current study, we also showed that PSII proteins such as PsbO and PsbA were more abundant in CCC than in the PCC chloroplasts. Since PSII proteins including PsbO and PsbA accumulate in the grana stack of thylakoid membranes, there should be a good correlation between the relative amount of PSII proteins and grana. This is further supported by ultrastructural analysis showing that the granal index approximately 50% more grana were present in the CCC than in the PCC chloroplasts in closely related single-cell C_4_ species *B. cycloptera* [3]. In addition, PPDK and PsbO polypeptides showed different accumulation patterns even though they are both nuclear-encoded. Similarly, chloroplast-encoded RbcL and cytochrome f did not show any correlation with their distribution. These results indicate that the accumulation of each of these four polypeptides was controlled independently. This is also the case in Kranz-type C_4_ plants, in which each photosynthetic gene is expressed in a cell-specific manner mediated by its own regulatory machinery [24,34,42].

The *in situ* hybridization analysis showed that the accumulation of *rbcL*, *psaB*, and *psbA* transcripts appeared to be similar in the two types of chloroplasts in young and intermediate leaves, whereas they were more abundant in the CCC than in the PCC in mature leaves (Figure 6D–L). This is further supported by the quantitative real-time PCR results (Figure 7). A similar selective partitioning of *rbcL* transcript accumulation in the chloroplasts of fully differentiated cells in the single-cell C_4_ species was reported [32]. These results indicate that the expression of these chloroplastic genes appear to be regulated at the transcriptional level or mRNA stability during leaf development. In particular, the distribution of *rbcL* transcripts appeared to correlate with the RbcL protein distribution throughout leaf development, suggesting that the *rbcL* expression is predominantly controlled at these levels. The PsbO and PsbA immunolocalization result, which represents PSII distribution, indicated that the CCC chloroplasts contained approximately 50% more PSII than in the PCC chloroplasts. This is in agreement with the confocal quantitative analysis of PSII in the dimorphic chloroplasts of *Bienertia sinuspersici* showing that the CCC had a higher intensity of chloroplast fluorescence compared to that of the PCC [43]. As expected, the gene encoding subunits of photosystem I *psbA* transcripts appeared to accumulate excessively in the CCC chloroplasts. It is possible that the *psbA* transcripts, coding for the D1 reaction centre protein of PSII, are used for immediate translation of the D1 subunit for the repair of photodamaged PSII complexes occurs in the CCC chloroplasts that have a higher grana index. Similarly, *psbA* transcript accumulation was observed in chloroplasts of the resurrection plant, *Xerophyta humilis* [44]. We observed lower levels of *psaB* transcript in the CCC because PSI proteins accumulate more in agranal chloroplasts (as in PCC) than in grana-rich chloroplasts (as in CCC) in maize [29], and this correlates with PsaB polypeptide distribution. 

The mechanisms controlling the development of single-cell C_4_ photosynthesis in plants are not fully understood. Overall, transcript distribution analyses suggest that the rate of transcription of photosynthetic genes in the CCC chloroplasts appeared to generally be higher than that in the PCC regardless of the amount of polypeptides in the mature leaves of *B. sinuspersici*. Alternatively, chloroplastic transcripts could be degraded at a faster rate in the PCC than that in the CCC. The expression of chloroplastic genes is regulated primarily at the post-transcriptional level [45]. For example, *psbA* expression is mainly controlled at translational steps in green algae and higher plants [45,46]. While high levels of *psbA* transcripts accumulate, the synthesis of PsbA protein is strictly controlled by light and development [47,48,49,50,51,52,53]. Moreover, inhibition analysis using the translation inhibitor chloramphenicol shows an increase in mRNA level of *psaB* and *psbA*, while *rbcL* transcript levels are unaffected in *Chlamydomonas reinhardtii* [54]. From these observations and our results, it is possible that the expressions of *psaB* and *psbA* genes are possibly controlled at the translational level compared to that of *rbcL*. Recent studies identified transcription factors such as S1-domain RNA binding protein (RSLB), which interacts with *rbcL* mRNAs in *Amaranthus* [55], *Flaveria* [56], *Zea mays* [55,57], and *Arabidopsis* [55,57,58], and its polypeptide localization in the CCC of *Bienertia* was reported [43]. It is possible that these photosynthetic genes are likely controlled at multiple levels for fine-tuning protein accumulation to respond to environmental signals such as light and temperature. The observed changes in *rbcL* transcripts and polypeptide in chloroplasts of chlorenchyma cells at different stages during leaf development in this study function as a platform for future experiments to examine transcriptional and post-transcriptional mechanisms that regulate the expression of RbcL and other chloroplast-encoded genes in *B. sinuspersici*.

## 4. Materials and Methods

*Bienertia sinuspersici* was grown as previously described [33]. Briefly, *B. sinuspersici* was vegetatively propagated from shoot branches of healthy plants. After rooting, plants were grown in controlled environment chambers under 350 µmol m^−2^ s^−1^ light with day/night temperatures of 25/18 °C and 14 h/10 h photoperiod. Leaf samples from three- to four-month-old plants were used for all experiments.

Various developmental stages of leaves of *B. sinuspersici* were studied by analyzing the youngest (0.1 cm), young (0.3 cm), intermediate (0.5–0.6 cm), premature (1.0–1.2 cm), and mature (>2.0 cm) leaves. Three leaf samples of each developmental stage from at least three independent plants were used for structural and immunolocalization electron microscopic studies. Observations from previous studies showed that the cellular differentiation of chlorenchyma cells during leaf development occurs in a basipetal direction with undifferentiated cells at the base and differentiated ones at the tip [7,32]. Thus, samples (0.2 cm in length) were prepared in the middle region of leaves from young, intermediate, premature, or mature stage, whereas the whole leaves of the youngest were harvested and chemically fixed in 1.25% (*v*/*v*) glutaraldehyde and 2% (*v*/*v*) paraformaldehyde in 50 mM PIPES buffer (pH 7.2) containing 0.3 M mannitol at 4 °C overnight. The fixed leaves were washed with the same buffer, dehydrated through an ethanol series, and embedded in LR White resin (London Resin Company Ltd., Berkshire, UK). Sections (70 nm thick) were performed on an ultramicrotome and collected on 150 mesh nickel grids. Sections were treated with TBST-BSA containing 25 mM Tris-HCl (pH 7.5), 150 mM NaCl, 0.1% (*v*/*v*) Tween 20, and 1% (*w*/*v*) BSA for 1 h, and then incubated with anti-Rubisco large subunit (Agrisera, Vännäs, Sweden, 1:1000), anti-*Zea mays* PPDK (courtesy of Chris Chastain, 1:1000), anti-PsbO (courtesy of Marilyn Griffith, 1:1000), or anti-PsaB, PsbA, anti- *Zea mays* cytochrome f (Agrisera, 1:100) at 4 °C overnight. After washing with TBST-BSA, sections were incubated with 10 nm gold conjugated anti-rabbit IgG (Sigma-Aldrich, Oakville, Canada) for 1 h. Sections were sequentially washed with TBST-BSA, TBST, and H_2_O, and then stained with uranyl acetate and lead citrate. Images were taken from 25 cells from three bioreplicates using a Philips CM-10 transmission electron microscope (FEI Company, Hillsboro, OR, USA) at an accelerating voltage of 60 kV. Sections were also incubated with preimmune rabbit sera or the omission of the primary antibody as controls. The gold particles were counted in several chloroplasts in each image, and the density of gold particles per unit area (μm) was calculated using Image J (http://rsbweb.nih.gov/ij/ accessed on 17 October 2011). Very low background labelling was observed in the negative control in which sections were treated as described above by omitting the treatment with primary antibodies (data not shown).

Various constructs were made by subcloning specific DNA sequences into the pBluescript SK+ (Agilent Technologies Canada, Mississauga, Canada) vector for in vitro transcription. Nucleotide sequences of *RbcL*, *PsaB*, *PsbA*, and *16S* rRNA were amplified by PCR from *B. sinuspersici* leaf genomic DNA. The *RbcL* coding sequence was amplified using forward (5′- CGC GGA TCC ATG TCA CCA CAA ACA GAG ACT AAA -3′) and reverse (5′- CGC GTC GAC AAA TTT GAT TTC CTT CCA TAC CT) primers and inserted into the *Bam*HI-*Sal*I sites of pBluescript SK+. The *PsaB* coding sequence was amplified using forward (5′- CGC CTC GAG TTA ACC GAA TTT GCC CGA TG -3′) and reverse (5′- CGC CTC GAG TTA ACC GAA TTT GCC CGA TG -3′) primers and inserted into the *Eco*RI-*Xho*I sites of pBluescript SK+. The *PsbA* coding sequence was amplified using forward (5′- CGC TCT AGA TTA TCC ATT TGT AGA TGG AGC TT -3′) and reverse (5′- CGC GGA TCC ATG ACT GCA ATT TTA GAG AGA CG -3′) primers and inserted into the *Xba*I-*Bam*HI sites of pBluescript SK+. The *16S* rRNA partial sequence was amplified using forward (5′- ACA CTC GAG CCG CAC AAG CGG TGG AGC AT -3′) and reverse (5′- CGC GGA TCC GTG ATC CAG CCG CAC CTT -3′) primers and inserted into the *Bam*HI-*Xho*I sites of pBluescript SK+. All constructs were purified using the Plasmid DNA Minipreps Kit (BioBasic Inc., Markham, Canada) and verified by sequencing before in vitro transcription.

*In situ* hybridization was performed according to Langdale et al. (1987, 1988) [11,35] with modifications. Briefly, samples from intermediate, mature leaves and shoot tips were fixed in a solution containing 70% (*v*/*v*) ethanol and 30% (*v*/*v*) acetic acid. The fixed tissue was dehydrated through an ethanol series, exchanged with tert-butyl alcohol, and embedded in paraffin. Sections (10 μm thick) were made on a rotary microtome and mounted onto silanated slides (Dako, Carpinteria, CA, USA). Mounted sections were deparaffinized in xylene, rehydrated in a graded ethanol and water series, and treated with proteinase K to remove RNase. From the various pBluescript SK+ constructs, sense and anti-sense RNA probes were generated by in vitro transcription reaction using MAXIscript kit (Thermo Fisher Scientific, Waltham, MA, USA). The probes were labeled with digoxigenin (Roche, Manheim Germany) during in vitro transcription and then alkaline-hydrolyzed to small fragments (150–200 bp). Deparaffinized sections were prehybridized, hybridized with the labeled RNA probes, and washed. The washed sections were incubated in a blocking solution followed by incubation with alkaline phosphatase conjugated anti-digoxigenin serum (Roche, Manheim, Germany). After washing to remove a non-binding antibody, visualization of the hybridized RNA probes was performed using 4-nitroblue tetrazolium chloride and 5-bromo-4chloro-3-indolyl-phosphate. All samples were counter-stained with 0.02% (*w*/*v*) Safranin O to visualize cell structures unless otherwise specified. 

The isolation of protoplasts from the mature leaves of *B. sinuspersici* was performed according to Lung et al. (2011) [33]. Briefly, the isolated chlorenchyma cells of *B. sinuspersici* were incubated in an enzyme solution containing 1.5 % (*w*/*v*) cellulase Onozuka R10 (Yakult Honsha Co. Ltd., Tokyo, Japan) and 0.1% (*w*/*v*) BSA in cell-stabilizing (CS) solution [0.7 M sucrose, 25 mM HEPES-KOH (pH 6.5),5 mM KCl and 1mM CaCl_2_] for 4 h in the dark without shaking. A homogeneous population of protoplasts was collected by washing or floating them twice on top the CS medium followed by a 2 min centrifugation step at 100 g. The purification of dimorphic chloroplasts from isolated chlorenchyma protoplasts of *B. sinuspersici* was obtained with the hypo-osmotic shock method according to Lung et al. (2012) [59]. Total RNA was extracted from purified chloroplasts using TRIzol Reagent (Invitrogen, Burlington, Canada) according to the manufacturer’s instruction. Contaminating genomic DNA was removed using the DNA-free Kit (Ambion, Austin, TX, USA) according to the manufacturer’s instruction. The quantity and purity of the RNA were determined spectrophotometrically. The integrity of the RNA was assessed by agarose electrophoresis. Single-strand cDNA was synthesized from 2 μg total RNA using the Protoscript II RT-PCR Kit (New England BioLabs, Ipswich, USA) and random nonamer according to the manufacturer’s instructions. Primers for real-time quantitative PCR were designed using the Primer-BLAST program (National Center for Biotechnology Information; accessed June 27, 2011) to produce amplicons of 121 to 167 bp. Secondary structures of amplicons were predicted using Mfold [60] to avoid too many secondary structures. To determine relative expressions, *16S* rRNA was amplified using forward (5′- GGA GCT GGC CAT GCC CGA AG-3′) and reverse (5′- GGT GAT CCA GCC GCA CCT TCC-3′) primers. Rubisco large subunit (RbcL)-encoding sequence was amplified using forward (5′- TGT TCT GCC TGT TGC TTC GGG A-3′) and reverse (5′- CGG TGC ATT TCC CCA AGG GTG T-3′) primers. PsaB-encoding sequence was amplified using forward (5′- TCG CTA TTC CCG GAG CCA GAG G-3′) and reverse (5′- AGT TCC CGC TCC TTG GGA GGT-3′) primers. PsbA-encoding sequence was amplified using forward (5′- TGG AGG AGC AGC AAT GAA CGC T-3′) and reverse (5′- AGA CGC GAA AGC GAA AGC CT-3′) primers. Real-time quantitative PCR (RT-qPCR) was performed in CFX96^TM^ Real time System (BioRad, Hercules) using SsoFast^TM^ EvaGreen Supermix (BioRad, Hercules, CA, USA) according to the manufacturer’s instruction. Single amplicons for each primer were confirmed with a single peak in a melting curve and a single band on an agarose gel. Eight independent RNA samples were tested in duplicate for each primer set. Relative gene expression was calculated using the Pfaffl method [61]. 

## 5. Conclusions

Here, we showed the expression of various key photosynthetic genes in the developing leaves of *B. sinuspersici*. While single-cell type C_4_ species mimic Kranz-type C_4_ plants in many aspects, their regulatory mechanisms for protein compartmentation must be very different from those in Kranz-type C_4_ plants because of their unique C_4_ photosynthesis system. The high-resolution quantitative analysis of photosynthetic proteins and transcript localization analysis suggested that independent and multiple levels of regulation of photosynthetic genes control the mode of photosynthesis of this species developmentally and perhaps environmentally. Our findings would help further studies on regulatory mechanisms of the differential distribution of photosynthetic proteins in the dimorphic chloroplasts of the single-cell type C_4_ species.

## Figures and Tables

**Figure 1 plants-12-00077-f001:**
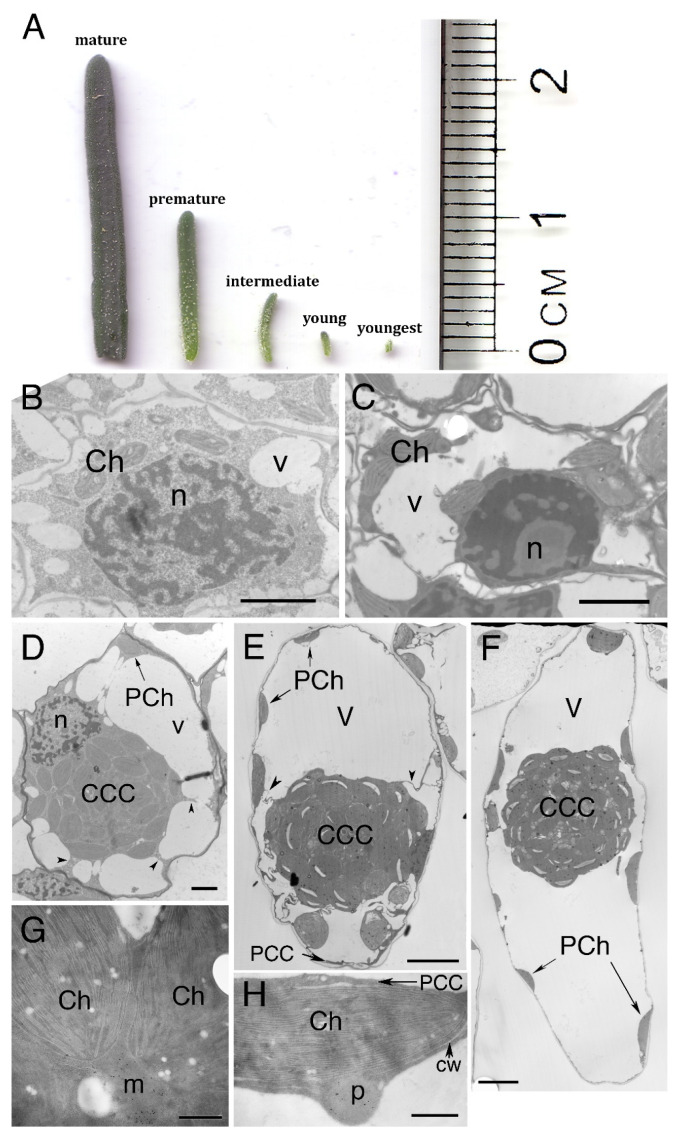
Transmission electron micrographs of chlorenchyma cells in developing leaves of *B. sinuspersici*. (**A**) Leaves were categorized into five developmental stages depending on their length: youngest (0.1 cm), young (0.2 cm), intermediate (0.5–0.6 cm), premature (1.0–1.2 cm), and mature (>2 cm). Transmission electron micrographs of chlorenchyma cells in (**B**) youngest, (**C**) young, (**D**) intermediate, (**E**) premature, and (**F**) mature leaves of *B. sinuspersici*. (**G**) Ultrastructure of chloroplasts in the CCC. (**H**) Ultrastructure of chloroplast in the PCC. Arrows, cytoplasmic strands; CCC, central cytoplasmic compartment; PCC, peripheral cytoplasmic compartment; Ch, chloroplast; cw, cell wall; m, mitochondrion; n, nucleus; p, peroxisome; PCh, PCC chloroplasts; v, vacuole; Bars = 2 μm in (**B**–**D**,**G**,**H**), 10 μm in (**E**,**F**).

**Figure 2 plants-12-00077-f002:**
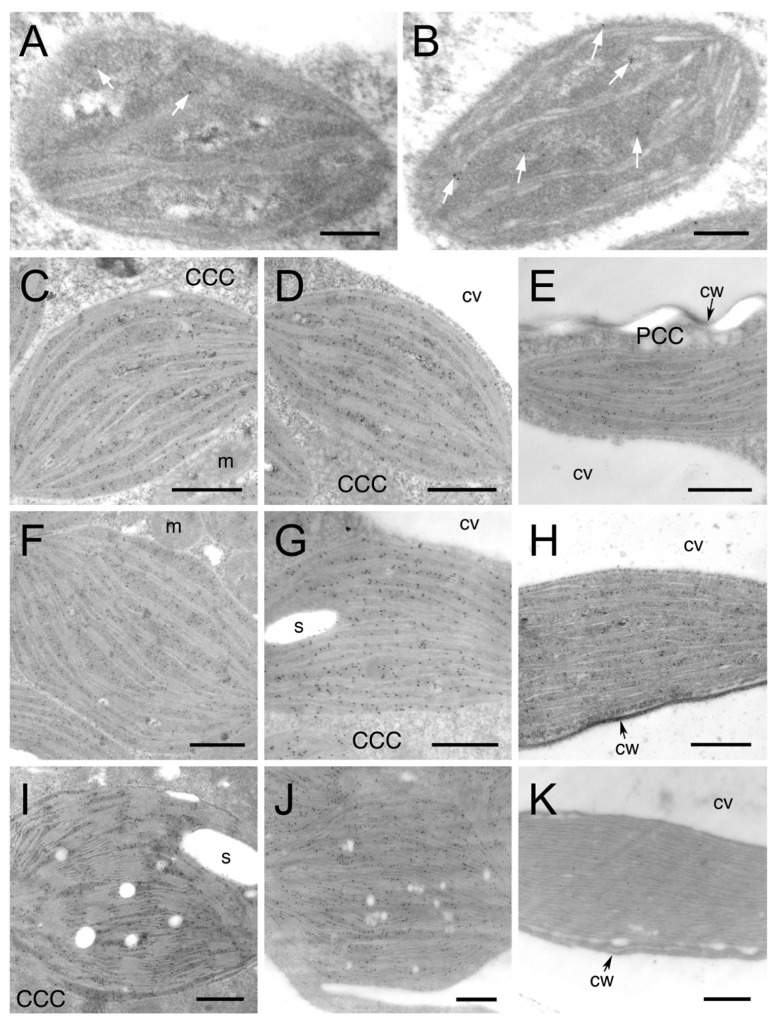
Immunolocalization of RbcL in chloroplasts in developing leaves of *B. sinuspersici*. Leaf cross-sections from various developmental stages were probed with RbcL antiserum and then with a gold-conjugated secondary antibody. Images are transmission electron micrographs of chloroplasts in (**A**) youngest, (**B**) young, (**C**–**E**) intermediate, (**F**–**H**) premature, and (**I**–**K**) mature leaves showing a specific reaction of the RbcL antibody. Starting from the intermediate stage, the CCC and PCC (**E**,**H**,**K**) are evident. Chloroplasts in the CCC were further divided into the inner layer of CCC (CCCI: **C**,**F**,**I**) and the outer layer of CCC (CCCO: **D**,**G**,**J**). White arrows indicate gold particles (**A**,**B**). RbcL, Rubisco large subunit; CCC, central cytoplasmic compartment; PCC, peripheral cytoplasmic compartment; s, starch; cv, central vacuole; cw, cell wall; m, mitochondria. Scale bars = 200 nm (**A**,**B**) or 500 nm (**C**–**K**).

**Figure 3 plants-12-00077-f003:**
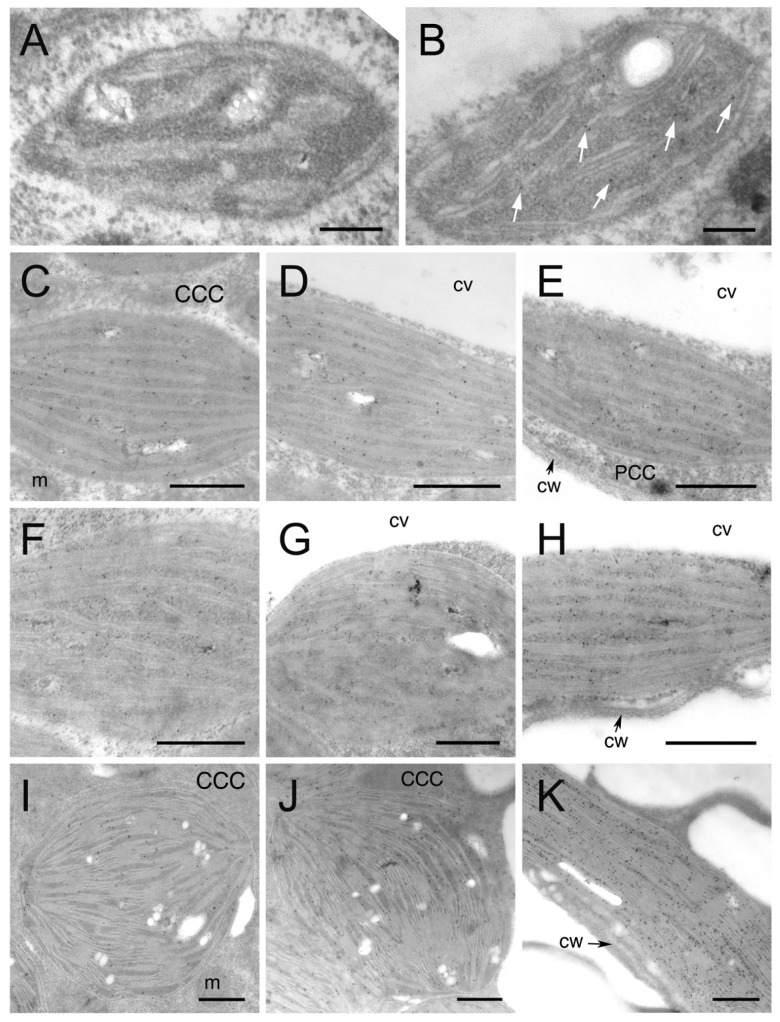
Immunolocalization of PPDK in chloroplasts in developing leaves of *B. sinuspersici*. Leaf cross-sections from various developmental stages were probed with PPDK antiserum and then with a gold-conjugated secondary antibody. Images are transmission electron micrographs of chloroplasts in (**A**) youngest, (**B**) young, (**C**–**E**) intermediate, (**F**–**H**) premature, and (**I**–**K**) mature leaves showing specific reaction of the PPDK antibody. Starting from the intermediate stage, the CCC and PCC (**E**,**H**,**K**) are evident. Chloroplasts in the CCC were further divided into the inner layer of CCC (CCCI: **C**,**F**,**I**), and outer layer of CCC (CCCO: **D**,**G**,**J**). White arrows indicate gold particles (**B**). CCC, central cytoplasmic compartment; PCC, peripheral cytoplasmic compartment; cv, central vacuole; cw, cell wall; m, mitochondria. Bars = 200 nm (**A**,**B**) or 500 nm (**C**–**K**).

**Figure 4 plants-12-00077-f004:**
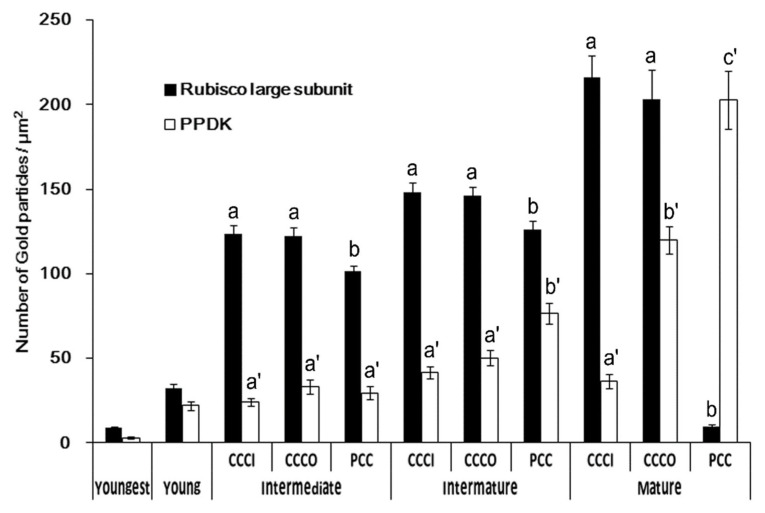
Quantification of RbcL and PPDK in chloroplasts in the developing leaves of *B. sinuspersici* based on immunolocalization analyses. The number of gold particles per unit area (μm^2^) was determined for RbcL and PPDK in developing leaves for chloroplasts in 25 cells from three biological replicates. Mean ± SE. Different letters represent a significant difference (a and b for RbcL; a’, b’, and c’ for PPDK) using the Student *t*-test (*p* < 0.01). RbcL, Rubisco large subunit; CCC, central cytoplasmic compartment; CCCI, CCC inner; CCCO, CCC outer; PCC, peripheral cytoplasmic compartment.

**Figure 5 plants-12-00077-f005:**
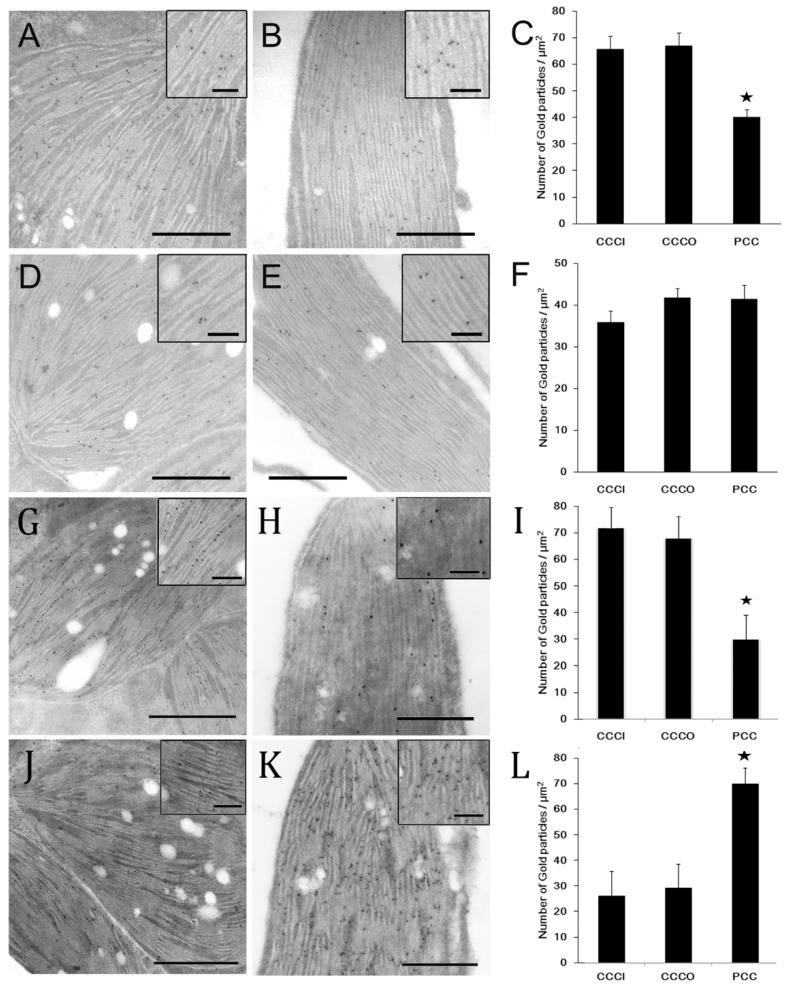
Immunolocalization and quantification of PsbO, cytochrome f, PsbA and PsaB in chloroplasts in mature leaves of *B. sinuspersici*. Immunolocalization and quantification of (**A**–**C**) PsbO, (**D**–**F**) cytochrome f, (**G**–**I**) PsbA, and (**J**–**L**) PsaB in chloroplasts of mature leaves. Images are transmission electron micrographs of (**A**,**D**,**G**,**J**) CCC and (**B**,**E**,**H**,**K**) PCC chloroplasts probed with PsbO, cytochrome f, PsbA, and PsaB antisera. Scale bars = 500 nm (100 nm for insets). Gold particles were counted and expressed as particle number per unit area (μm^2^) in chloroplasts in 25 cells from three different bioreplicates (**C**,**F**,**I**,**L**). Mean ± SE. Stars represent a significant difference using the Student’s *t*-test (*p* < 0.01). CCC, central cytoplasmic compartment; CCCI, CCC inner; CCCO, CCC outer; PCC, peripheral cytoplasmic compartment.

**Figure 6 plants-12-00077-f006:**
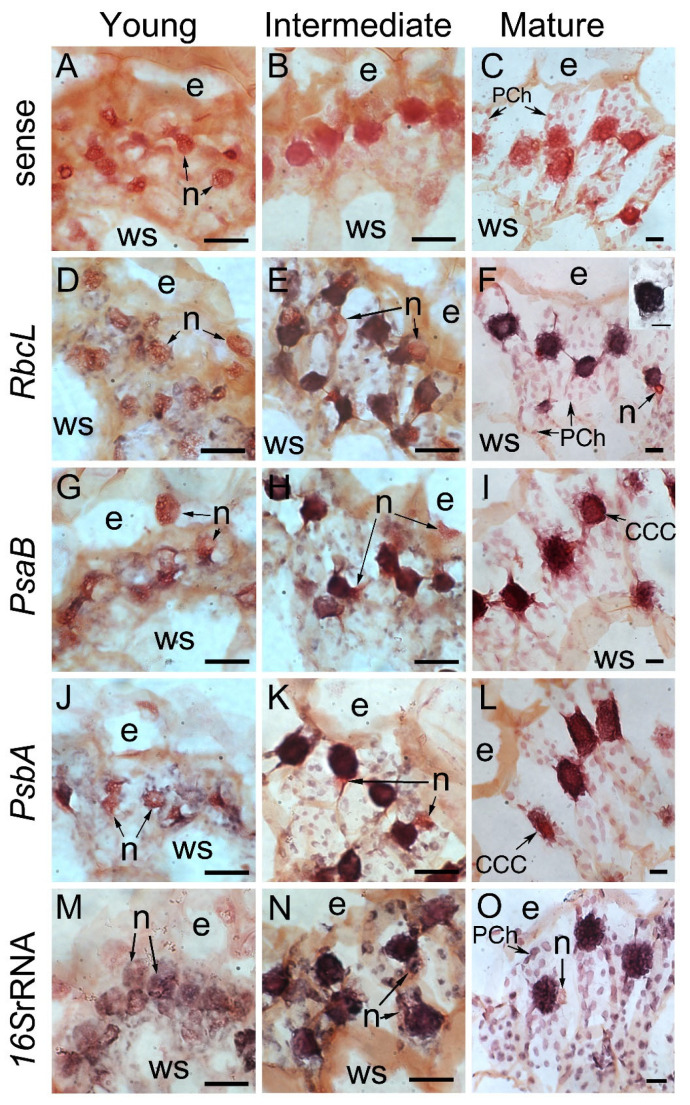
Chloroplastic transcript localization in chlorenchyma cells of developing *B. sinuspersici* leaves. Sections were prepared from shoot tips and mature leaves. Images are enlarged images of leaves at three developmental stages: (**A**,**D**,**G**,**J**,**M**) young, (**B**,**E**,**H**,**K**,**N**) intermediate, and (**C**,**F**,**I**,**L**,**O**) mature. Cross-sections were hybridized with labelled (**A**–**C**) sense *rbcL* or (**D**–**F**) antisense *rbcL*, (**G**–**I**) *psaB*, (**J**–**L**) *psbA*, or (**M**–**O**) *16S* RNA probes. Sections were counter-stained with Safranin O for the better visualization of cellular structures except for the inset image in (**F**). Individual chlorenchyma cells are outlined by a dashed line (**A**–**C**). Specific hybridization is shown in purple. CCC, central cytoplasmic compartment; PCh, chloroplasts in the peripheral cytoplasmic compartment (PCC); e, epidermal cells; ws, water storage cells; n, nuclei; *rbcL*, Rubisco large subunit. Scale bars = 20 μm.

**Figure 7 plants-12-00077-f007:**
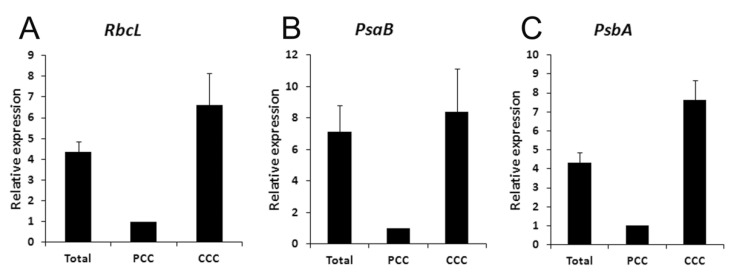
Real-time qPCR relative quantification of chloroplastic transcripts in two types of chloroplasts in mature leaves of *B. sinuspersici*. Chloroplastic RNAs isolated from purified total, PCC, or CCC chloroplasts were used as a template and PCR amplified using gene-specific primers for (**A**) *rbcL*, (**B**) *psaB*, and (**C**) *psbA* as described in Methods. The means of the results from duplicates of eight independent RNA templates are shown. Each result was normalized with threshold cycle (Ct) values of *16S* rRNA. Error bars indicate SE. *rbcL*, Rubisco large subunit; PCC, peripheral cytoplasmic compartment; CCC, central cytoplasmic compartment.

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
