# Peer review of "Development of C4 Biochemistry and Change in Expression of Markers for Photosystems I and II in the Single-Cell C4 Species, Bienertia sinuspersici"

_plants, 2022, doi:10.3390/plants12010077_

Round 1

Reviewer 1 Report

The two authors intend to contribute with their studies to improved understanding of the metabolism and expression for C4 photosynthesis and carbon assimilation in the single cell C4 species Bienertia sinuspersici. Their insights in the C4 metabolism based on studies during the plant development by in situ immune localization analysis through transmission electron microscopy and by in situ hybridization assays.

The paper is written in clear and comprehensible manner. The results were clearly presented and provide the basis for comprehensible and plausible explanations. The experiments are clearly described and the hypotheses and conclusions are traceable. However, to its best, the results presented can be acknowledged to be confirmative. And it is sensible to ask the authors for the novelty of their findings.

For non-expert of the C4 photosynthesis in Bienertia, the figures showing microscopic images of the cells in different developmental stages, are not self-explanatory.  How can the authors inform the reader that the chloroplasts belong to a particular stage of leaf development, as seen in Figures 1 and 2? What would be a control for the correctness of these data presented in the figures? As the possible differences in immune localization are not automatically be detected, it seems that it is not predictable, form which developmental state the chloroplasts are derived.

Figure 5 seems to me to be even more critical. It cannot be proven and shown without further control what the authors claim in the result section. It seems that actually everything and nothing at all can be deduced or interpreted from the figures without further representations, controls and additional explanations.

As only initial results by in situ hybridization and immune localization are provided, the authors naturally limit themselves to speculations and hypotheses regarding transcriptional and posttranscriptional regulation in the discussion. I am feel the intensive speculations are hardly based and justified by their experiments.  Even with a benevolent view the assumptions are not always convincing and a more cautious discussion is recommended.

Reviewer 2 Report

The paper by Yanagisawa and Chuong investigates the development of C4 photosynthesis in the terrestrial plant Binertia sinuspersici, a single-cell C4 photosynthesis type. The experimental approach include TEM immunolocalization of chloroplastic RbcL and PPDK, as markers for the equivalents to bundle sheath and mesophyll compartments (CCC and PCC, respectively), and in situ hybridization for chloroplast-encoding photosynthetic genes. This work contributes to increasing knowledge about the development of this type of C4 photosynthesis throughout leaf development. Although it also provides evidence of the existence of various regulatory mechanisms, the work is fundamentally descriptive and does not investigate these mechanisms.

The authors show that PSII is more abundant in chloroplasts of the central compartment whereas PSI is higher in those of the peripheral compartment. In Kranz-type C4 photosynthesis, PSII is more abundant in mesophyll cells. Explain this discrepancy.

Minor questions

-Title: Change to: “Development of C4 biochemistry and changes in the expression….”

-Revise the citation in text at the beginning of Introduction

-Show CCC and PCC in Figures

-Show s, cv, cw in Figure 1 and 2

-Show PCh in Figure 5

Round 2

Reviewer 1 Report

I would like to acknowledge the authors' efforts to heed the reviewer's comments. An additional introductory paragraph was included at the beginning of the result section, but they did not follow up the reviewer’s suggestions to indicate the leaf area which is used for the electron micrographs. It must be obvious that the chlorenchyma cells during leaf development differ in the top and the bottom part of leaves at any developmental stage, in particular when leaves are up to 2 cm long. Moreover, I cannot see in the discussion what the authors promised: “We have modified the discussion to be less intensive as recommended.” As we received the revised version of the manuscript, no changes were found in the discussion. In addition: The reply: “thus have added a new figure (Figure 1) to illustrate the development of chlorenchyma cells in various developmental stages of Bienertia sinuspersici leaves” is certainly an overstatement when figure panels from the supplements were transferred to the main body.  

Author Response

We thank the reviewer for their positive comments and insightful suggestions. We have incorporated the changes in the revised manuscript in an attempt to address the specific comments raised by the reviewer. We hope that these revisions alleviate all of the reviewer’s concerns.

Round 2 Comment:

I would like to acknowledge the authors' efforts to heed the reviewer's comments. An additional introductory paragraph was included at the beginning of the result section, but they did not follow up the reviewer’s suggestions to indicate the leaf area which is used for the electron micrographs. It must be obvious that the chlorenchyma cells during leaf development differ in the top and the bottom part of leaves at any developmental stage, in particular when leaves are up to 2 cm long. Moreover, I cannot see in the discussion what the authors promised: “We have modified the discussion to be less intensive as recommended.” As we received the revised version of the manuscript, no changes were found in the discussion. In addition: The reply: “thus have added a new figure (Figure 1) to illustrate the development of chlorenchyma cells in various developmental stages of Bienertia sinuspersici leaves” is certainly an overstatement when figure panels from the supplements were transferred to the main body.

Response: As recommended, we added a few statements in the Materials and Methods section that describe the where about the leaf samples were obtained from the various developmental stages. As suggested, we have taken the advice of the reviewer and have made numerous changes to the Discussion to the best of our abilities without losing the important message by adding some new references to substantiate our observations and removing some indirect evidence and placing less emphasis on overstatements that were made in the discussion to address concerns raised by the reviewer